# Can Marketing Increase Willingness to Pay for Welfare-Enhanced Chicken Meat? Evidence from Experimental Auctions

**DOI:** 10.3390/ani13213367

**Published:** 2023-10-30

**Authors:** Lenka van Riemsdijk, Paul T. M. Ingenbleek, Hans C. M. van Trijp, Gerrita van der Veen

**Affiliations:** 1Research Centre Digital Business and Media, University of Applied Sciences Utrecht, 3584 BK Utrecht, The Netherlands; gerrita.vanderveen@hu.nl; 2Marketing and Consumer Behaviour Group, Wageningen University, 6708 PB Wageningen, The Netherlands; paul.ingenbleek@wur.nl (P.T.M.I.); hans.vantrijp@wur.nl (H.C.M.v.T.)

**Keywords:** animal-friendly products, certified labels, consumers, marketing, positioning strategies, real-life experiment, willingness to pay

## Abstract

**Simple Summary:**

Consumer concern for animal welfare is currently not fully reflected in the market share of animal-friendly products. Marketing strategies for animal-friendly products typically emphasize sustainability-related benefits, such as animal welfare, while existing research suggests that consumers prioritize personally relevant benefits, such as taste and curiosity. This study tests the effectiveness of positioning strategies emphasizing personally relevant benefits, namely curiosity, in a real-life experiment at the point of purchase, also measuring the effects of certified labels and the impact of consumer attitudes towards eating meat. It conducts experimental auctions with 101 Dutch university students and measures their willingness to pay (WTP) for a lunch meal with chicken meat. Results indicate that both the positioning strategy and the certified label significantly increase consumer WTP, with the highest WTP generated when both elements are present (without providing evidence for an interaction effect). This implies that to maximize sales of welfare-enhanced meat companies should combine positioning strategies that emphasize personally relevant benefits with certified labels that can support the claimed animal friendliness. Since our results also suggest that consumers with conflicting feelings towards meat are less sensitive to such strategies, some care should be taken when designing awareness campaigns about the negative effects of meat consumption.

**Abstract:**

Consumer concern for animal welfare is currently not fully reflected in the market share of welfare-enhanced meat. A possible solution is developing marketing strategies that emphasize personally relevant benefits such as taste and curiosity, instead of having a sole focus on sustainability-related benefits, since existing research indicates that the former are more appealing to most consumers. This study tests strategies positioning welfare-enhanced meat as personally relevant in a real-life experiment and how consumer attitudes towards eating meat influence reactions to the positioning strategies. The study conducts experimental auctions with 101 Dutch university students, manipulating the positioning strategy and a certified animal welfare label and measuring participants’ willingness to pay (WTP) for a lunch meal with chicken meat. Results indicate that all manipulations significantly increase consumer WTP, with higher WTP for certified labels than for the positioning strategy, and the highest WTP for the combination of both elements (without providing evidence for an interaction effect). This implies that companies should combine positioning strategies that emphasize personally relevant benefits with certified labels. Since the effectiveness of such strategies may be limited for consumers with conflicting feelings towards meat, some care should be taken when designing awareness campaigns about the effects of meat consumption.

## 1. Introduction

Understanding what drives consumers to purchase animal-friendly products is crucial to expand the market of welfare-enhanced meat [1]. So far, companies selling welfare-enhanced meat seem to struggle in finding effective marketing strategies that stimulate consumers to switch from conventional meat to welfare-enhanced meat [2,3]. Numerous surveys indicate that consumers find animal welfare important, that they look for labels that help them identify animal-friendly products and that they are willing to pay for products with higher animal welfare [4,5,6,7]. Despite these positive beliefs and attitudes, consumers still mainly opt for conventional meat instead of meat produced with higher animal welfare standards, such as free-range or organic [3,8]. In the Netherlands, for example, the market share in 2017 was only 14% for welfare-enhanced beef and 19% for welfare-enhanced poultry [9], whereas 85% of Dutch consumers expressed that they are willing to pay more for animal-friendly products [6]. Companies selling welfare-enhanced meat could therefore greatly benefit from research studying how marketing can help translate consumer concern and attitudes towards animal welfare into actual purchase behaviour.

At the same time, the effectiveness of marketing strategies for animal-friendly products is hindered by consumer attitudes. Negative events and information like the horsemeat scandal from 2013 are likely to create negative feelings about meat, which may lead to conflicting or ambivalent attitudes towards meat [10,11]. Ambivalent attitudes mean that consumers have simultaneously positive (e.g., tasty, high nutritional value) and negative beliefs about meat (e.g., unhealthy, causes animal suffering). Ambivalence has been found to weaken the translation of positive consumer attitudes to purchase intentions [11], thus possibly presenting a challenge to marketing welfare-enhanced meat.

The existing literature has identified two crucial elements of marketing strategies that increase consumer preference for welfare-enhanced meat and, possibly, other food products. First, positioning strategies make animal-friendly products appealing and attractive through emphasizing benefits such as taste or curiosity, instead of having emphasis only on the product’s animal friendliness [12,13]. This is necessary for a majority of consumers who find animal welfare important, but still prioritize personally relevant benefits, such as taste, health and price [14,15,16]. Second, reliable certification is of particular importance to create trust in animal welfare claims because consumers cannot verify such claims themselves [17,18].

An important methodological limitation of the body of literature on marketing strategies for animal-friendly products is that empirical studies testing the effects of marketing instruments on consumer decisions commonly made use of experimental stimuli that placed respondents in hypothetical situations (so-called vignette studies) [19]. Because participants do not experience the consequences of their answers, these methods are more likely to induce participants to give socially desirable answers, particularly in the case of product attributes with a social dimension, such as animal welfare [20]. The socially desirable answers may result in overstating the importance of animal welfare, also known as social desirability bias [20]. The absence of real consequences in hypothetical situations also leads to an over-estimation of their real willingness to pay (WTP) for animal-friendly products, also known as hypothetical bias [5,21,22]. While previous research gives encouraging evidence that positioning strategies can increase consumer welfare-enhanced meat choice in hypothetical situations [12], we still know very little about whether such strategies could actually make consumers willing to pay more for welfare-enhanced meat at the point of purchase.

This study aims to address this gap by testing the effects of marketing strategies which use product positioning and certified labels in a more realistic context. By using a non-hypothetical context, where consumers actually have to pay for animal-friendly products, this study helps to overcome hypothetical bias and can therefore help reveal whether or not marketing strategies increase consumer WTP for welfare-enhanced meat at the point of purchase. As a second contribution, the present study investigates the role of consumer ambivalence towards meat, i.e., the extent to which the consumer holds both positive and negative feelings towards eating meat [11], in the effectiveness of the marketing strategies on consumer WTP. We show that consumers with moderately ambivalent feelings about eating meat show less consistent behaviour, in that their positive perceptions of welfare-enhanced meat lead to a lower marginal WTP for such meat as compared to consumers with non-ambivalent feelings.

In the remainder of this article, we introduce our conceptual framework and hypotheses by first reviewing consumers’ perceptions of animal-friendly products, followed by discussing how marketing strategies can influence consumer perceptions and, thus, increase consumer WTP for animal-friendly products. Next, we discuss the theoretical foundation of ambivalence towards meat and explain how experimental auctions can help accurately measure consumer WTP for welfare-enhanced meat, and how we used experimental auctions to test the hypothesized relationships. We conclude with the discussion of the results and a research agenda for companies selling welfare-enhanced meat.

## 2. Theoretical Background

### 2.1. Understanding Consumer Perceptions of Animal-Friendly Products

When designing appealing marketing strategies, marketers need to understand consumer motives and perceptions of animal-friendly products. Existing research has generally shown that while consumers find animal welfare important, the majority of consumers still prioritize personally relevant benefits, such as taste, health, product safety and price [14,15,16]. Animal-friendly food choice in general, and meat choice in particular, typically presents consumers with a so-called social dilemma because they must trade off animal welfare for other product benefits, such as price and (perceived) availability [1]. A social dilemma reflects a situation when the (product) choice that maximizes one’s short-term individual welfare negatively impacts long-term societal welfare [23] and it is believed to be a major barrier for consumers to buy animal-friendly and other ethical products [24,25]. To address this barrier, manufacturers of animal-friendly products (as well as policy makers or animal-interest organisations) may design campaigns that reinforce animal welfare with personally relevant benefits. In the Netherlands, where higher animal welfare standards are reflected in higher prices of animal-based food products, welfare-enhanced meat can, for example, be positioned as healthier and tastier [26].

Because consumers differ in their perceptions, attitudes and behaviour towards animal-friendly products, a number of studies have distinguished different consumer segments [1,27,28]. These studies have shown that socio-demographic factors, such as gender, education or the presence of children, may partially explain the differences between consumer segments [27,28,29]. Psychographic factors, such as values and beliefs in relation to animals, consumer lifestyles or personality characteristics are however suggested as more powerful explanations of these differences because they are closer related to choice behaviour [1,30].

### 2.2. Marketing Strategies for Animal-Friendly Products

The marketing strategy refers to companies’ decisions that have a major impact on creating, communicating, delivering and exchanging value with the companies’ customers and other stakeholders [31], and it typically includes decisions pertaining to segmentation, target-market selection and product positioning [32]. The literature dedicated to studying which marketing strategies effectively stimulate consumer animal-friendly product choice has identified several marketing strategies which are particularly suitable for animal-friendly and other ethical products [13,33,34,35]. These strategies typically focus on the following issues: enhancing consumer opportunity, i.e., by providing a broad and easily available assortment of animal-friendly products; facilitating consumer ability, i.e., by increasing awareness about animal welfare and providing trustworthy information through certified labels; and facilitating consumer motivation, i.e., by making animal-friendly products appealing and attractive through product positioning strategies [13,36].

Rather than considering each strategy to be an independent element, our research views each strategy to be a cornerstone necessary to increase the sales of animal-friendly products. First, marketers need to use positioning strategies to make animal-friendly products appealing and attractive through emphasizing personally relevant benefits such as taste or curiosity [12,13] to attract the majority of consumers who find animal welfare important, but still prioritize personally relevant benefits [14,15,16]. Second, marketers need to use reliable certification to create trust in animal-welfare claims because consumers cannot verify such claims themselves [17,18]. Good availability and broad assortment are then preconditions for consumer purchase of animal-friendly products.

### 2.3. Ambivalence towards Meat

Consumer perceptions and attitudes towards meat are greatly influenced by public information and events, such as the horsemeat scandal, studies on the carcinogenicity of red and processed meat, and accusations of companies repackaging meat past its sell-by date. It is likely that such events create negative feelings about meat and, for some consumers, this may lead to conflicting attitudes towards meat [10]. In the literature, such conflicting feelings or attitudes are referred to as ambivalence towards meat [11]. The extent to which a consumer believes that eating meat has benefits in terms of, for example, nutritional value and tastiness, as well as disadvantages in terms of, for example, unhealthiness, environmental problems and the moral aspects of killing animals, defines how much a consumer is ambivalent towards meat [10].

Ambivalence has been found to influence the effect of attitudes on behavioural intentions in consumer meat choice [11], also referred to as the meat paradox [37]. Consumers with higher levels of ambivalence, i.e., those who hold positive as well as negative feelings towards eating meat, showed a weaker relationship between their attitudes and intentions related to meat choice [11], reduced meat consumption [10] and intentions to reduce their future meat consumption [10]. The literature distinguishes between two types of ambivalence towards meat: latent and felt ambivalence [10]. While latent ambivalence assumes the existence of positive as well as negative evaluations in one’s memory [38], it can lead to feelings of discomfort when brought to one’s attention in a decision-making context. This feeling of discomfort is conceptualized as felt ambivalence [39].

Understanding how ambivalence influences consumer behaviour is important for at least two reasons. First, it may help increase consumer welfare-enhanced meat choice and/or decrease overall meat consumption. Second, it may help companies selling welfare-enhanced meat to segment their customer base. Despite its potential, ambivalence has hardly been studied in the current literature [37].

### 2.4. Willingness to Pay

Willingness to pay refers to the price premium that an individual is willing to pay to obtain a wanted benefit, such as animal welfare, or to avoid an unwanted characteristic, such as unhealthy ingredients [5,40]. Consumers’ WTP is thus a measure of consumer purchase behaviour and it is believed to reflect the total perceived value of the product [41,42]. An accurate appraisal of consumer WTP for animal welfare is critical to effectively market animal-friendly products in that it helps to develop new products, design promotional strategies and set up pricing tactics [43,44]. Studies measuring consumer WTP for animal welfare indicated a small positive WTP for animal welfare (for a review, see [5]) which varies by a number of factors, such as animal type, product type and region. For example, ref. [5] concluded that consumers are willing to pay most for the welfare of beef cows, and the least for the welfare of pigs; and that consumers in Southern Europe are willing to pay more for animal welfare than consumers in Northern Europe.

An accurate estimation of consumer WTP has been an important objective of many marketing studies [44], which has logically resulted in a huge variety of techniques and methods that are used to test consumer WTP (for a review, see [43]). The techniques can be classified along several dimensions, of which we discuss the three most important ones. First, while some methods, such as market observations or experiments, use revealed preferences, others, such as surveys, use stated preferences [43]. Second, the stated preference methods can either directly ask respondents to indicate their WTP, or they can indirectly estimate consumer WTP from their rankings or ratings of different products [43]. Finally, while some methods measure consumers’ hypothetical WTP, other use non-hypothetical (also called actual or incentive-aligned) WTP, in which participants are obliged to purchase the product if they claim that they are willing to pay the price [44]. As we will explain later, we will use experimental auctions in this study, which measure non-hypothetical WTP, using direct measurement of stated preferences.

While existing research shows that, generally, consumers are willing to pay for animal welfare (for a review, see [5]) it also shows that the WTP significantly varies across welfare levels (cage-free, free-range or organic) [45] and geographical regions [46]. Moreover, it provides little guidance for companies on how to market their animal-friendly products to increase consumer WTP. The few exceptions are studies measuring how consumer WTP for animal welfare increases when supported by a certified label [20,47] or studies linking (comparing) WTP for animal welfare to (WTP for) other product attributes, such as taste and food safety (e.g., [48,49]). To advance our understanding on how marketing can increase consumer WTP for animal welfare, the present study tests the hypothesized relationships that we present in the next section.

## 3. Conceptual Framework and Hypotheses

The conceptual framework of this study corresponds to the two aims of this study. First, it draws relationships between marketing strategy and WTP for welfare-enhanced meat (H1, H2). Second, it places ambivalence towards meat in the framework, and to do so it also includes customer value as a concept (H3, H4).

### 3.1. Increasing Consumer WTP with Marketing Strategies

Consumers’ WTP is seen in the consumer behaviour literature as a measure of a broader construct that reflects consumer purchase behaviour [43]. WTP is defined as the price premium that a consumer is willing to pay to obtain a certain benefit, such as taste, good feeling or animal welfare [5,40]. A logical starting point for marketers who want to increase consumer WTP for their products is therefore to identify which benefits consumers find the most important when buying animal-based food. Consistently, existing research finds that the majority of consumers prioritize personally relevant benefits, such as taste, health, quality and safety [1,14,15,16]. Animal welfare and other sustainability-related benefits are also important for most consumers, but they are not prioritized over personally relevant benefits [1]. This means that consumers are willing to pay more for personally relevant benefits such as taste and healthiness than for animal welfare. If consumers thus believe that animal-friendly food offers personally relevant benefits, this will translate into WTP for these products. To communicate personally relevant benefits, marketers can use positioning strategies that emphasize these benefits, for example, by claiming that animal-friendly dairy is tastier than regular dairy or that welfare-enhanced meat is of higher quality than regular meat. Thus far, to our knowledge, no research has tested whether such positioning claims increase consumer WTP, but previous research [12,50] suggests that such effects may exist. We thus hypothesize:

**Hypothesis** **1** **(H1).**
*Positioning strategy increases consumer WTP for welfare-enhanced meat.*


Next to the positioning strategies, stakeholder endorsement may be another powerful tool that can increase consumer WTP for welfare-enhanced meat. In the food category, stakeholder endorsement is typically communicated with a third-party certified label, such as an animal welfare label (e.g., Animal Welfare Approved), a fair-trade label (e.g., Max Havelaar) or a general quality label (e.g., EU CE marking). Research has shown that certified labels can increase the trustworthiness of the product and its claims [51,52]. While certified labels can potentially guarantee the claimed benefit related to a wide range of product attributes, they are particularly valuable for so-called credence attributes that consumers cannot verify themselves, as opposed to search or experience attributes [47,53]. By making consumers certain that the product offers the claimed benefits, certified labels are likely to increase the overall value perceptions of the product, which may encourage product purchase.

The value of animal welfare certified labels has been studied for several product categories, for example meat, fish and dairy [20,47]. These studies show that certified labels endorsing the products’ animal friendliness are a potentially powerful tool to increase consumer WTP for animal-friendly products. In the present study, stakeholder endorsement will be communicated with a trustworthy animal welfare label, so we expect a similar effect.

**Hypothesis** **2** **(H2).**
*Stakeholder endorsement increases consumer WTP for welfare-enhanced meat.*


### 3.2. Total Perceived Value and Ambivalence towards Meat

To test the effect of ambivalence, we include total perceived value in our model. If companies want to persuade consumers to buy animal-friendly products, they need to increase the total perceived value of such products (sometimes called utility) [54]. Total perceived value has been found to predict a number of important inter-related behavioural outcomes, such as purchase intentions, consumer product choice, consumer WTP or consumer word-of-mouth [54,55]. Total perceived value is a multidimensional construct that includes various types of product benefits in relation to the perceived costs [56]. The existing literature proposed various taxonomies of total perceived value (for reviews, see [56,57]), which generally depend on product category and/or purchase situation [57]. For welfare-enhanced meat, the total perceived value can be classified into two groups of particular relevance: individualistic value and ethical value. This classification reflects the social dilemma typical for welfare-enhanced meat choice, in which consumers must trade off their individual welfare for animal welfare [24,25]. Hence, consumers who wish to maximize the ethical value, i.e., the capacity to contribute to the improvement of public welfare in general, and animal welfare in particular [58], must give up (some of) the individualistic value which serves their own welfare. For welfare-enhanced meat, the individualistic value typically includes the following sub-categories [12]: functional, which refers to the functional quality and performance; emotional, which refers to the product’s capacity to arouse consumer feelings; monetary, which refers to the value for money; and epistemic, which refers to a product’s capacity to arouse curiosity [57,59].

Total perceived value has been recognized as the key driver of consumer purchase behaviour in general [60,61], and animal-friendly purchase behaviour in particular [1,12,62]. With regards to consumer WTP as a specific measure of consumer purchase behaviour, authors generally agree that total perceived value is an important predictor of consumer WTP [41,42,63]. (Some studies, e.g., [64], however, suggest that the relationship between total perceived value is mediated by other constructs, such as customer satisfaction.) We thus hypothesize:

**Hypothesis** **3** **(H3).**
*Total perceived value has a positive effect on consumer WTP for welfare-enhanced meat.*


Ambivalence towards meat refers to the existence of conflicting attitudes or feelings towards meat [11]. Ambivalence towards meat is caused by the existence of negative issues stemming from meat consumption, such as moral issues due to the suffering of animals, ecological issues and health issues, and at the same time the existence of positive effects of eating meat, such as sensory pleasure and tradition [37]. Consumers who are highly ambivalent hold positive as well as negative feelings towards meat, while those who are not ambivalent (prevailingly) hold only one type of feeling [11]. Consumers who prevailingly hold negative feelings towards meat are typically non-omnivores, who thus experience no inner conflict because they do not eat meat [37]. Consumers who prevailingly hold positive feelings towards meat also arguably experience no inner conflict because their meat choice is driven by positive attitudes towards meat. Consumers with highly ambivalent feelings, however, typically experience an inner conflict also called the meat paradox [37]. The meat paradox essentially refers to a weak, or inconsistent, effect of ambivalent consumer attitudes on consumer behaviour. In the context of product choice, this effect has been observed as a relatively weak effect of consumers’ attitudes towards the product on consumer purchase intentions [11]. In other words, [11] have found that consumers with higher levels of ambivalence showed a weaker relationship between their attitudes and intentions related to meat choice than consumers with lower levels of ambivalence. Since total perceived value can be viewed as a measure of consumer attitudes and WTP as a measure of purchase intention [42,65], we expect ambivalence to have a similar effect on these constructs. Specifically, we expect that ambivalence towards meat will weaken the effect of consumers’ value perceptions on their WTP for welfare-enhanced meat. We thus hypothesize:

**Hypothesis** **4** **(H4).**
*Ambivalence towards meat moderates the effect of total perceived value on consumer WTP for welfare-enhanced meat, so the less the consumer is ambivalent towards meat, the stronger the effect.*


## 4. Materials and Methods

Experimental auctions are among the most popular methods to measure consumer WTP [5] and they have been increasingly popular to measure WTP for credence attributes, such as animal welfare [66]. One of the main advantages of experimental auctions is that they are incentive-aligned, where real money is exchanged for actual products, so they encourage participants to bid exactly their WTP [67]. Research suggests that incentive-aligned methods are preferred to non-incentive-aligned (hypothetical) methods because they make participants more price sensitive [44]. By using real money, which consumers have to pay for products on the basis of their answers, experimental auctions help decrease hypothetical bias because consumers are less likely to give socially desirable answers [22].

Originally, experimental auctions used the procedure called the Vickrey nth-price auction [68,69]. In the Vickrey nth-price auction, group sessions are used to make participants compete for a product, as the product is typically sold to the highest bidder. This procedure is different to actual purchase behaviour, such as when consumers shop for food, as there is typically sufficient supply for all shoppers rather than just one or a few products being available to the highest bidders [68]. Moreover, if consumers are confronted with the bids of others and have to compete for a product, their competitiveness can distort their WTP. Therefore, the lottery procedure developed by [70] (BDM) is more suitable for simulating actual shopping for food [68].

In the BDM procedure, similarly to the Vickrey auctions, each participant submits a sealed bid for the product(s). Rather than selling the product to the highest bidder, in the BDM procedure the actual price to be paid is randomly determined, and all participants who bid at or above the actual price (are allowed to) buy the product for its actual price. The BDM procedure therefore encourages participants to bid exactly their WTP [68] and generates the most similar WTP (compared to open-ended questions, choice-based conjoint and incentive-aligned choice-based conjoint) to the real purchase data [44]. Because BDM generates WTP very similar to the real purchase data and its procedure is comparable to actual food purchases, we used the BDM procedure in our study.

### 4.1. Design

The hypothesized relationships were tested in a non-hypothetical experiment (*N* = 101) conducted in the Netherlands. The participants were presented with lunch meals with chicken meat, specifically wraps with crispy chicken meat, as these present a common lunch meal for the participants in our sample. Chicken meat was selected because it is, together with beef and pork, one of the types of meat most consumed by Dutch consumers [71,72]. Moreover, while certain consumers may avoid eating beef or pork due to their religion, most non-vegetarian consumers typically eat chicken meat.

We manipulated the positioning strategy (PS) and certified label (CL) of the chicken meat in a 2 × 2 within-subjects design, and also included a reference product. This resulted in 5 products: reference, control (no PS/no CL), positioned (yes PS/no CL), certified (no PS/yes CL) and certified positioned (yes PS/yes CL). Figure 1 provides an overview of the different lunch meals.

### 4.2. Sample

124 adult consumers who eat chicken meat participated in the experiment. Data were collected during 4 weeks in November and December 2018 at Wageningen University (NL) and the participants were recruited by the researchers involved in this study. The participants voluntarily signed up online, via e-mail or face to face and received a lunch at the University canteen for their participation. Since the aim of the study was to measure participants’ WTP for improved animal welfare, participants had to be familiar enough with the situation of social dilemma that is characteristic for animal-friendly food choice [1,73]. We therefore restricted our study to participants who had lived in the Netherlands for at least 3 years, resulting in a sample of 101 participants. Almost all participants (99%) were University students. The mean age of the participants was 22 years, ranging from 18 to 32 years, and 28% were men.

### 4.3. Stimuli

The participants were presented with five products, each complemented with a short product description (see Figure 1 for all product descriptions). First product, labelled as regular wrap, contained a regular chicken meat with no improved animal welfare. The price of the regular wrap was EUR 3.50, equal to the price of the crispy chicken wrap sold at the university canteen at the time of the experiment. Next to the regular wrap, participants were presented with four alternative wraps (A, B, C and D), which all contained a welfare-enhanced (free-range) chicken meat of Dutch origin. The alternative wraps were manipulated in a 2 × 2 design, which manipulated positioning strategy (yes/no) and certified animal welfare label (yes/no).

The positioning strategy aimed to increase the epistemic value, i.e., curiosity, since curiosity is one of the influential motives driving welfare-enhanced meat choice [12,74]. Positioning strategy thus emphasized that the chicken had lived in an innovative, free-range welfare-enhanced husbandry system which created a natural environment for the animals, provoking curiosity in the consumer. The selected certified label is also known as the better-life label, issued by The Dutch Society for the Protection of Animals, which is considered a well-known and trustworthy label by Dutch consumers [75]. The certified label showed two out of a maximum three better-life stars, which typically refers to free-range husbandry systems for farm animals.

Next to the explanation of the manipulated product attributes (certified label and/or innovative husbandry system), the information presented with alternatives A, B, C and D included the free-range character and the Dutch origin of the meat, but excluded the product price, as we aimed to measure consumer WTP for these products. The order of the alternative wraps was changed several times to eliminate order effects [76]. A pre-test study (*N* = 10) helped to ensure that the stimuli, the rules of the experimental auctions and the questions in the questionnaire were clear.

### 4.4. The BDM Procedure

Experimental auctions using the BDM procedure [70] were organized to collect data as the BDM procedure best simulates shopping for food [68]. Participants were invited to join a study measuring how people make choices when buying food. The experiment took 25 min on average and the participants received a EUR 7.50 voucher to buy lunch at the university canteen after finishing the experiment. While the participants made their choices independently, several participants could participate at the same time.

First, it was explained to participants that their answers would have actual consequences on the type of products they would receive as part of their lunch and it was therefore in their best interest to be absolutely honest.

Second, we explained the rules of experimental auctions. One rule was that they would get a pre-specified lunch meal as part of their lunch. Another rule was that they could exchange the regular lunch meal for an alternative if they were willing to pay the actual price for the alternative product. The price of the alternative product would only be known to them at the end of the experiment, so they were encouraged to write down their exact WTP for each alternative. We also explained that, at the end, only one alternative product was going to be available (next to the regular product), so they must compare each alternative product to the regular product rather than the different alternative products to each other. This prevented participants choosing one alternative product, and providing their WTP for this alternative only, so we could obtain participants’ WTP for all alternatives.

Third, to ensure that participants understood the rules and the procedure, a practice round with another product type (orange juice) was used. Participants were asked to write down how much extra they would be willing to pay for an alternative orange juice (fair-trade labelled or in a bottle from recycled plastic). We then revealed that only the fair-trade alternative was available, and randomly determined the price for this alternative. Participants were then told which orange juice they would get (regular or fair-trade) if they were satisfied with their product and the price they had to pay. After having answered participant’s questions, the experimental auction with the lunch meals could start.

Fourth, participants were told that they would receive wraps with crispy chicken meat as part of their lunch. They were given the opportunity to exchange the regular wrap for an alternative wrap (A, B, C and D) if they were willing to pay the actual price for the alternative wrap. They could inspect all alternatives and were encouraged to carefully read the product descriptions. Fifth, participants wrote down their WTP for all alternative wraps. Sixth, they completed a pen-and-paper questionnaire.

Finally, at the end of each experiment, participants randomly drew the actual price for the available alternative wrap, which was the certified free-range wrap. If participant’s WTP for the certified wrap was at or above the actual price, he/she got the certified wrap for the actual price, and the remaining value on a voucher to buy products of his/her choice. Otherwise, the participant got the regular wrap and the remaining value (EUR 4) to buy other products.

### 4.5. Questionnaire

The questionnaire began with questions about participant’s past animal-friendly shopping behaviour with regards to different categories of animal-based food (meat, eggs, milk), internal reference price for a regular crispy chicken meat wrap, questions measuring participant’s value perceptions of the wrap with certified positioning, ambivalence towards meat, attitudes towards animal welfare, meat consumption, familiarity with the better-life certified label and concluded with classification questions.

#### Measures

Total perceived value is a formative construct [77], with dimensions depending on the type of product [78]. For welfare-enhanced meat, the relevant dimensions are functional (tastiness), emotional (good feeling), ethical (animal welfare) and epistemic value (curiosity) [12,74,79]. Monetary value, which refers to the value for money, was not included because the products were presented without the prices, and participants could thus not evaluate the value for money. Moreover, the effect of monetary value on consumer WTP is arguably different than those of other value perceptions, since when consumers evaluate a product as offering good value for money, they may not be willing to pay more for this product (see also [80]). The role of social value is unclear, as some studies (e.g., [58], tested with organic dairy) conclude that social acceptance is influential in predicting consumer animal-friendly purchase intentions, while others (e.g., [12] tested with free-range meat) show that it is not. As part of the social value is captured in the ethical value construct, in that contribution to animal welfare is considered to be a socially desirable behaviour [81], we did not include social value as a separate construct. Total perceived value was measured with five items adapted from [12,82], of which functional, emotional and epistemic value were measured with one item each and ethical value with two items to achieve a more balanced ratio between individualistic and ethical value. Participants compared two wraps, namely the control and the certified positioned, on a 7-point scale, where the lowest value refers to the control wrap being much better, while the highest value refers to the certified positioned wrap being much better. The total perceived value was calculated as the average of the five items.

Ambivalence towards meat was measured with 5 items adapted from [10]. The items measured felt ambivalence with 3 questions and 2 statements on a 7-point scale, where the lowest value refers to participant feeling no ambivalence towards meat, while the highest value refers to participant feeling maximum ambivalence, i.e., conflicting feelings, towards meat. Since our study confronted participants with various types of meat which differed in their level of animal welfare, felt ambivalence rather than latent ambivalence was a more suitable measure (see also [10]). The ambivalence towards meat was calculated as the average of the five items. Reliability analysis showed that the five items yielded a Cronbach’s alpha of 0.884, suggesting that the items constituted a reliable measure of ambivalence towards meat.

## 5. Results

### 5.1. Willingness to Pay for Animal Wefare

To examine the effects of the positioning strategy and certified label, we performed a detailed analysis of consumer WTP for the welfare-enhanced wraps (see Figure 2 for the mean values and Figure 3 for sample distribution). We could observe a small positive WTP for animal welfare only, M = EUR 0.14, 95% CI [0.11, 0.17], which translates to a 4% price premium that, on average, consumers were willing to pay on top of the regular wrap, which cost EUR 3.50. As our data further show, half of the participants were willing to pay a maximum of EUR 0.10, and almost 40% of our sample were not willing to pay extra for animal welfare.

The positioning strategy added more value to the welfare-enhanced meat, with an average WTP of EUR 0.27, 95% CI [0.22, 0.31], which translates to a 7.6% price premium that, on average, consumers were willing to pay on top of the regular wrap. Additional analyses show that almost 25% of the participants were not willing to pay extra for the positioning strategy, and that half of them were willing to pay a maximum of EUR 0.20.

The certified label elicited a slightly higher WTP than the positioning strategy, M = EUR 0.34, 95% CI [0.29, 0.38], which translates to a 9,6% price premium that, on average, consumers were willing to pay on top of the regular wrap. Only 12% of our sample reported not being willing to pay extra for the certified label, and half of the participants were willing to pay a maximum of EUR 0.30.

Finally, the certified positioning yielded the highest WTP, M = EUR 0.46, 95% CI [0.40, 0.52], which translates to a 13.2% premium that consumers, on average, were willing to pay for a wrap with certified free-range chicken meat produced in an innovative husbandry system compared to a wrap with conventional chicken meat. The certified positioning yielded the highest median value of all manipulations, EUR 0.40, and 93% of our sample were willing to pay extra for the product with certified positioning.

### 5.2. Hypotheses Testing

Given the structure of our data set (with repeated measures data on WTP for different conditions and total perceived value for only one condition), we used two different analysis techniques to test our hypotheses. A repeated-measures ANOVA was used to test the effects of the positioning strategy (H1) and stakeholder endorsement (H2) and their interaction on consumer WTP for welfare-enhanced meat in our within-subjects design. A simple linear regression was used to analyse the effect of total perceived value (H3) as well as the interaction between total perceived value and ambivalence towards meat (H4) on WTP for welfare-enhanced meat, using standardised scores for each variable. All analyses were run in SPSS Statistics 23 and their results are presented in Table 1.

Hypothesis 1 predicted that positioning strategy would have a positive effect on consumer WTP for welfare-enhanced meat. Our results support this hypothesis, as a repeated-measures ANOVA, with Pillai’s trace, showed that positioning strategy significantly increased consumer WTP for welfare-enhanced meat (V = 0.574, F(1, 100) = 135.000, *p* < 0.001).

Hypothesis 2 predicted that stakeholder endorsement would have a positive effect on consumer WTP for welfare-enhanced meat. Our results support this hypothesis, showing that stakeholder endorsement significantly increased consumer WTP for welfare-enhanced meat (V = 0.332, F(1, 100) = 49.643, *p* < 0.001).

Hypothesis 3 predicted that total perceived value would have a positive effect on consumer WTP for welfare-enhanced meat. A simple linear regression, including total perceived value, ambivalence towards meat and their interaction in the model, showed that total perceived value significantly increased consumer WTP for welfare-enhanced meat (b = 0.207, *p* = 0.027), thus supporting Hypothesis 3.

Hypothesis 4 predicted that ambivalence towards meat would moderate the effect of total perceived value on consumer WTP for welfare-enhanced meat, so the less ambivalent a consumer was towards meat, the stronger the effect. Our results show that there was a significant interaction effect (b = −0.192, *p* = 0.035) on consumer WTP for welfare-enhanced meat. This provides support for Hypothesis 4, showing that ambivalence towards meat moderated the effect of total perceived value on consumer WTP for welfare-enhanced meat in that the more ambivalent the consumer was towards meat, the weaker the effect. To further explore whether this effect occurred for all values of ambivalence, we performed a floodlight analysis as advised by [83] that identified Johnson–Neyman significance regions between which the effect was significant at *p* = 0.05 (see Figure 4). These significance regions refer to values between 1 and 3.62, on a 7-point scale where the lowest values (1) refer to no ambivalence at all and the highest value (7) to maximum ambivalence. Overall, our results thus provide partial support for Hypothesis 4 in that ambivalence towards meat attenuated the effect of total perceived value on consumer WTP for welfare-enhanced meat, but this effect was restricted to consumers who experienced no to moderate ambivalence towards meat.

Finally, we also tested whether the positioning strategy interacted with stakeholder endorsement, such that, when used together, their effect on consumer WTP for welfare-enhanced meat was stronger than when used individually. Our results did not support this hypothesis, showing no significant interaction effect (V = 0.001, F(1, 100) = 0.072, *p* = 0.789). As shown in Figure 2, although the combination of positioning strategy and stakeholder endorsement generated the highest WTP, this presented a direct cumulative effect of both elements rather than an interaction effect. We return to these findings in the Section 6.

## 6. Discussion

This study investigated how marketing strategies using product positioning and stakeholder endorsement increase consumer WTP for welfare-enhanced meat, and how consumer ambivalence towards meat interacts with consumer value perceptions in influencing consumer WTP. Our results reveal that both a positioning strategy that arouses curiosity and an animal welfare certified label directly increase consumer WTP. Consumers are willing to pay EUR 0.20 on average (almost 6%) for a certified label and EUR 0.13 (almost 4%) for a new, innovative husbandry system (on top of the 4% that they are willing to pay extra for animal welfare). Furthermore, our findings suggest that ambivalence towards meat can present a challenge to increasing consumer welfare-enhanced meat choice. Our results show that consumers with moderately ambivalent feelings about eating meat show less consistent behaviour in that their positive perceptions of welfare-enhanced meat lead to a lower marginal WTP for such meat as compared to consumers with non-ambivalent feelings.

If we compare our findings on consumer WTP for animal welfare (i.e., the WTP for the free-range meat without a marketing strategy), the results show some consistency with recent results from a special Eurobarometer, a large-scale European survey with more than 27 thousand participants [6]. According to our data, 62% of consumers were willing to pay extra for welfare-enhanced meat (Eurobarometer reports 59% as the EU average, but 85% in the Netherlands), with 37% reporting a maximum of 6% price premium (Eurobarometer reports that 35% of EU/Dutch consumers are willing to pay 5% extra). The differences can be explained by using different methodologies (Eurobarometer measured hypothetical WTP), population (Eurobarometer used a representative sample) and specificity (Eurobarometer measured WTP for non-specified animal-friendly products, and with non-specified elements, such as certified labels). Our study used experimental auctions where participants had to pay for the products. This approach helped to minimize hypothetical bias, as participants are likely to give honest answers if their answers have consequences on the products that they will consume and the price that they must pay for these products. In that respect, our findings confirm that consumers are likely to pay more for animal-friendly products, but, importantly, they need encouragement in the form of positioning strategies and certified labels to really do so. This conclusion leads to important implications.

Finally, while we did find that positioning strategy and stakeholder endorsement jointly accumulated the highest levels of WTP, we did not find evidence for an interaction effect between the two variables. These findings imply that the two variables constitute different effects. The stakeholder-endorsement effect in that respect is most likely explained by the trust that consumers have in the communication instruments (in particular, welfare labels) that guarantee higher levels of animal welfare. This is comparable to effects previously found in the literature (cf. [17,18]). The effect of positioning strategy comes on top of that, suggesting that it is a different reason for consumers to pay more for the product than the improved welfare itself. The most likely explanation then is that the way the product is positioned indeed raises curiosity and that this curiosity is something that represents value to the consumer. This finding is in line with a previous study that found that so-called epistemic value is an important source for consumers to derive value from in animal-friendly products [12].

## 7. Implications

For policy makers, the findings imply that WTP for animal welfare is unlikely to emerge all by itself. As consumers need some help from marketers to pay more for animal-friendly products, policy makers should engage in partnerships with retailers and brand producers to materialize the latent demand for animal-friendly products. One concrete action that policy makers can carry out to encourage companies to invest in the marketing of their products is to conduct a large national-level survey that identifies the market segments and their associated WTP for animal-friendly products if they were supported by positioning strategies and certified labels. This would reduce the risks for companies associated with investments in their marketing strategies. In countries that lack a strong infrastructure for animal welfare certification, policy makers may also develop the organization for certified labels so that the industry can build on labels that are trusted by the general public.

Our results also provide companies with valuable and reliable insights on how to market their welfare-enhanced meat, i.e., which elements of marketing strategy drive consumers to pay the highest price. They suggest that companies may use each element on its own considering its unique contribution to consumer WTP. Essentially, the results of the current study are encouraging for the companies selling welfare-enhanced meat, since they show that even small changes, such as adding a certified label, can considerably increase consumer WTP. To boost consumer WTP even further, companies could combine different strategies. This implies that to maximize sales of welfare-enhanced meat, companies may combine positioning strategies that emphasize appealing product benefits, such as curiosity, with certified labels that can support the claimed animal friendliness. However, also in countries where animal welfare labels are absent, companies can already increase the WTP for animal-friendly products if they position their products more as personally relevant.

The increasing ambivalence towards meat may present a challenge to companies selling welfare-enhanced meat. Currently, NGOs and the media, but also governments, create campaigns that aim to encourage consumers to decrease their meat consumption, i.e., by emphasizing negative issues associated with meat, such as unhealthiness, animal welfare and environmental consequences. These campaigns, while certainly addressing an important issue, may also increase consumer ambivalence towards meat, which translates into weak relationships between perceived value and WTP (and arguably purchase intention) for meat. This suggests that even when companies manage to position welfare-enhanced meat as offering a higher perceived value, the increase in value will not fully translate into consumer WTP for such meat, eventually pushing the prices of welfare-enhanced meat downward. Since consumers with ambivalent feelings are arguably those who are highly concerned about animal welfare (yet unlikely to abandon all meat products from their diets), who are the target market for welfare-enhanced meat, this may present a threat to the growth of the market for welfare-enhanced meat. The solution to this problem is not straightforward, since governments are likely to further promote healthy diets which include eating less meat. As the first step, however, governments could investigate the side effects, e.g., in terms of discouraging consumers from buying welfare-enhanced meat, in the development of future meat campaigns.

An important remaining question is whether the price premium that consumers are willing to pay is proportional to the additional costs of animal-friendly production systems. Existing research on animal welfare economics estimated that animal-friendly production can cost from as little as 5% extra, to as much as 50% extra, depending on the product type and animal welfare level, among other factors [84]. For example, while the minimum additional costs of improved pig welfare from conventional systems to free-range systems only present an increase of 4–8%, the minimum additional costs of organic systems are 31% [85]. Taking a different approach, ref. [84] conducted market-level quantitative assessments, i.e., estimations based on scenarios where markets fully switch to animal-friendly systems, with the majority (80%) of producers upgrading to moderate animal welfare systems, and the rest (20%) to high welfare systems. Ref. [84] concluded that covering increased animal welfare standards would, for example, require a 36% increase in pork and an 8% increase in beef prices in the Netherlands. Overall, these numbers are larger than the consumer WTP for welfare-enhanced meat, which means that companies need to look for marketing strategies that can increase consumer WTP for welfare-enhanced meat.

## 8. Limitations and Future Research

Our findings should be seen in the light of their limitations. First, while the research design of experimental auctions makes an important contribution to the external validity of consumer animal-welfare research, the sample that we used cannot be taken as generalizable. In that respect our study provides more insights on the process through which marketing strategies influence consumer WTP for animal welfare. Future research may therefore complement our findings in larger representative country samples that can give more accurate estimates of how much consumers on a country level are willing to pay for animal-friendly products that are supported by marketing strategies. Second, our study included ambivalence towards meat as an individual difference variable, but did not include other consumer personal characteristics, such as values [86] and thinking style [87]. In larger country samples such variables may be added because they provide potentially valuable information for marketers that aim to identify and describe consumer segments with distinctive responses to marketing instruments and (related to that) distinctive levels of WTP. Finally, we tested our theory on one meat and meal type only, namely chicken meat used in a wrap. Future research may examine the generalizability of our findings by applying our method to other meat and meal types.

## 9. Conclusions

This study revealed that increasing the attractiveness of animal-friendly products, either by emphasizing personally relevant benefits in the positioning strategies, or by adding a certified label, can significantly increase consumer WTP. It also showed that consumers with conflicting feelings towards meat may be less sensitive to marketing strategies that increase the attractiveness of animal-friendly products, suggesting that some care should be taken when designing awareness campaigns about the negative effects of meat consumption.

## Figures and Tables

**Figure 1 animals-13-03367-f001:**
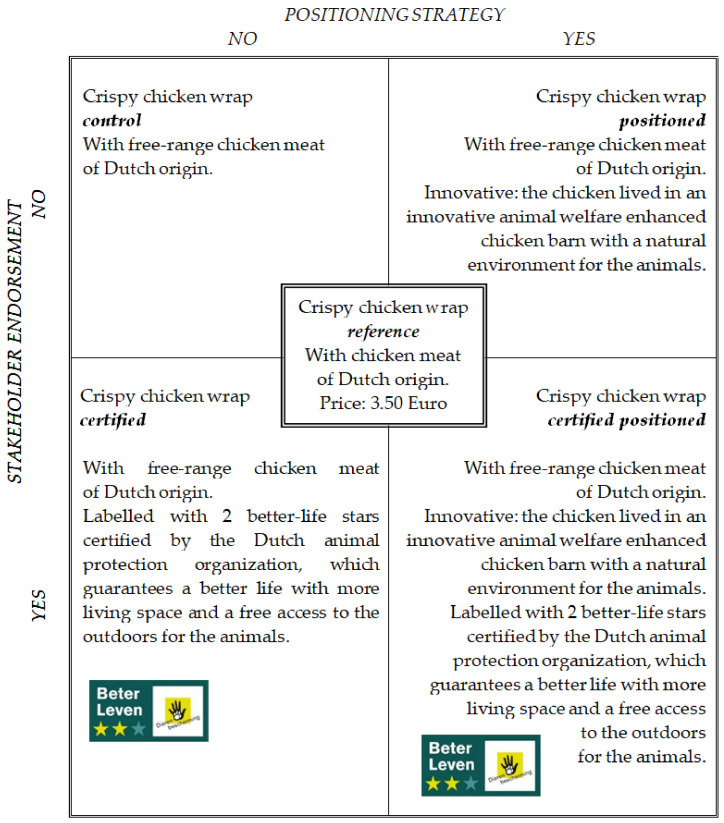
Experimental study design indicating all experimental stimuli.

**Figure 2 animals-13-03367-f002:**
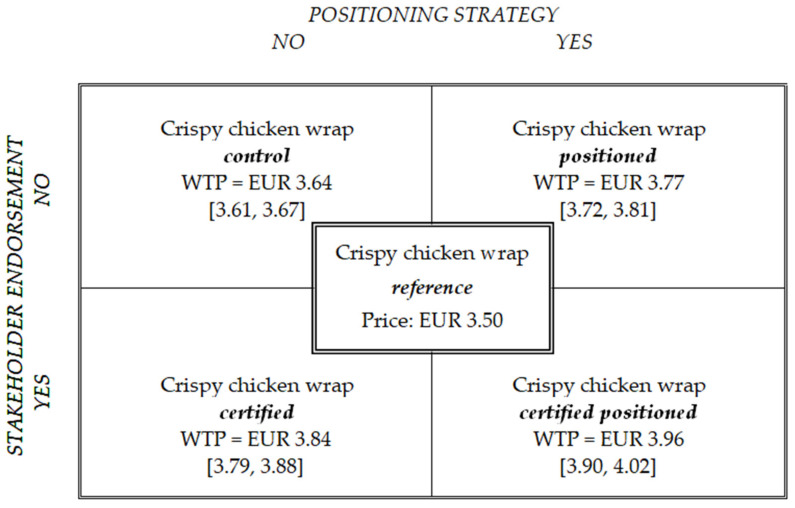
Results of willingness to pay for welfare-enhanced wraps. Note: WTP refers to mean total willingness to pay for the product (*N* = 101). Values in square brackets represent 95% confidence intervals.

**Figure 3 animals-13-03367-f003:**
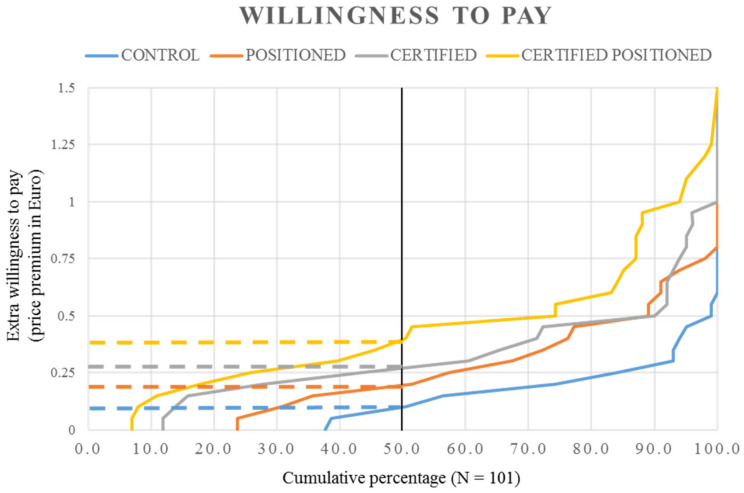
Distribution of participants’ willingness to pay for animal welfare-enhanced wraps.

**Figure 4 animals-13-03367-f004:**
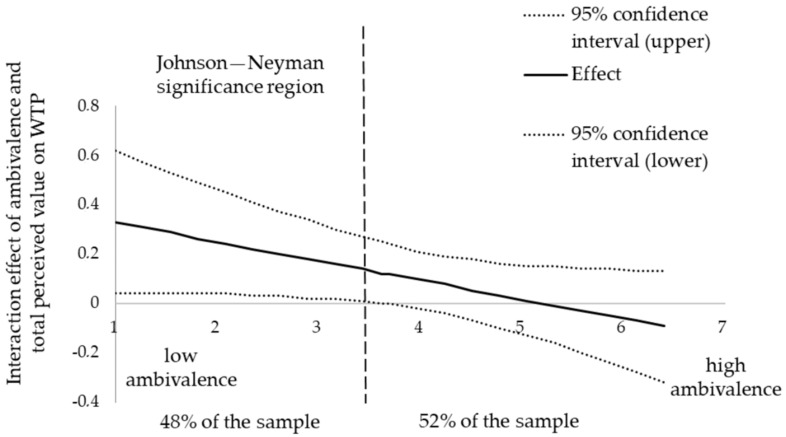
The interaction effect of ambivalence towards meat and total perceived value on consumer WTP for welfare-enhanced meat.

**Table 1 animals-13-03367-t001:** Parameters and effect sizes of hypothesized relationships.

Relationship		Parameter	Significance ^a^
Repeated-measures ANOVA		F-statistics (Pillai’s trace)	
H1	Positioning strategy → WTP for welfare-enhanced meat	0.574	0.000
H2	Stakeholder endorsement → WTP for welfare-enhanced meat	0.332	0.000
	Positioning strategy × stakeholder endorsement → WTP for welfare-enhanced meat	0.001	0.789
Model F(1, 100) = 249.28, *p* < 0.001			
Simple linear regression		b	
H3	Total perceived value → WTP for welfare-enhanced meat	0.207	0.027 ^a^
	Ambivalence → WTP for welfare-enhanced meat	0.048	0.637
H4	Total perceived value * ambivalence → WTP for welfare-enhanced meat	−0.192	0.035 ^a^
Model F(3, 97) = 2.27, *p* = 0.085			

^a^ one-tailed significance values.

## Data Availability

The data presented in this study are available upon request from the corresponding author.

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
