# Peer review of "Can Marketing Increase Willingness to Pay for Welfare-Enhanced Chicken Meat? Evidence from Experimental Auctions"

_animals, 2023, doi:10.3390/ani13213367_

Round 1

Reviewer 1 Report

Report on the manuscript animals-2564749 entitled: Can marketing increase willingness to pay for animal welfare enhanced meat? Evidence from experimental auctions.

I am a newbie regarding 'experimental auctions’, so I read the manuscript with huge interest.

After reading the title, the abstract and the abstract, one got the sense that it might be an insightful article. Then, the introduction, the theoretical framework and the description of the hypotheses with such exhaustive bibliographic justification just blow you away.

Nevertheless, one must not forget that this is a 'research article' not a 'thesis'. So, one wonders: were the last 365 lines necessary? Were they unnecessary? Could they not have been summarized?

However, once you get to M&M and R&D it can be read that the authors only worked with ‘crispy chicken wraps’ and that, after having done such extensive work in the previous sections, the discussion of the results is disappointing.

Some comments:

-         The title is neither accurate nor precise. The authors only considered 'chicken wraps' so they do not know what would happen in the case of other species or if the sample had been offered in another format. For example, what would happen if a whole chicken thigh or wing were offered? Etc.

-         The authors use 'animal welfare-enhanced meat'. I wonder whether 'animal' is a necessary term since 'welfare' and 'meat' are also in the same phrase.

-         The authors do not describe the parameters/variables that define 'welfare'. Were the UE requirements regarding broilers considered? Were the consumers informed about the new laws regarding broilers and laying hens?

-         The bibliographic work is exhaustive. However, they describe and justify their study over many pages and then spend less than half a page discussing their results.

For the manuscript to be accepted for publication, the authors must shorten the first 365 pages and improve the discussion.

They must be more accurate and precise. Only chicken wraps were considered so it is not adequate to talk with such liberty about ‘meat’.

-          L. 14. ‘testS’.

-          L. 16. ‘the purchase point’ or ‘the point of purchase’.

-          L. 20. ‘the highest…’.

-          L. 21. ‘welfare-enhanced…’.

-          L. 27 and 30. ‘welfare-enhanced meat’.

-          L. 36. ‘labelS’ and ‘for the combination of…’

Reviewer 2 Report

The paper reads well. My concerns are mainly related to methods.

Round 2

Reviewer 1 Report

The authors have responded very adequately to the reviewer.

There are some grammatical, bibliographic formatting and typographical errors that will surely be solved later.

--

Reviewer 2 Report

My comments has been sufficiently address. Thanks!